# Process Optimization and Modeling of Methylene Blue Adsorption Using Zero-Valent Iron Nanoparticles Synthesized from Sweet Lime Pulp

**Naincy Sahu [1], Shalu Rawat [1], Jiwan Singh [1],\*, Rama Rao Karri [2],\* , Suhyun Lee [3], Jong-Soo Choi [3] and Janardhan Reddy Koduru [3],\***

[1] Department of Environmental Science, Babasaheb Bhimrao Ambedkar University, Lucknow 226025, India; Naincysahu.29@gmail.com (N.S.); Shalurawat200@gmail.com (S.R.)
[2] Faculty of Engineering, University Teknologi Brunei, Jalan Tungku Link, Mukim Gadong A BE1410, Brunei Darussalam
[3] Department of Environmental Engineering, Kwangwoon University, Seoul 139-701, Korea; minisu87@naver.com (S.L.); spiruna03@naver.com (J.-S.C.)
\* Correspondence: jiwansingh95@gmail.com (J.S.); kramarao.iitd@gmail.com (R.R.K.); reddyjchem@gmail.com (J.R.K.); Tel.: +82-02-940-5496 (J.R.K.)

**Abstract:** The presence of dyes in waterbodies poses severe problems in human and aquatic creatures, and the development of treatment methods for the removal of these pollutants is of utmost importance. This research study investigates the elimination of methylene blue (MB) from an aqueous solution using zero-valent iron nanoparticles synthesized from sweet lime pulp waste (nZVISLP). The purity, chemical composition, and crystalline size of nZVISLP were investigated using microscopic and spectroscopic studies. A maximum MB removal efficiency of 98.9% was obtained at the following optimal conditions: $C_0$: 10 mg/L, dosage: 1.2 g/L, and temperature: 25 °C. To understand the adsorptive removal characteristics of nZVISLP, the investigational adsorption data were tested with conventional kinetic and isotherm models. Furthermore, a differential evolution optimization (DEO) technique was used to estimate the optimal intrinsic parameters in the isotherm and kinetic models. For the various evaluated isotherms, the correlation coefficient ($R^2$) values for the Freundlich and Sips isotherm models were ~0.98, thus confirming the aptness of these isotherms to represent MB adsorption onto nZVISLP. The robustness of non-linear models was verified by statistical metrics, thus validating the performance of the optimization technique. The results derived from this study affirm the potential of an ecofriendly biogenic nanomaterial, nZVISLP, for MB adsorptive removal.

**Keywords:** adsorption; methylene blue; zero-valent iron nanoparticles; differential evolution optimization; process modeling; sweet lime pulp waste

## 1. Introduction

Dyes and pigments are found to be major water pollutants that contaminate surface and groundwater severely, and several of these synthetic dyes show mutagenic and toxic effects on human health. Dyes have a complex molecular structure that makes it difficult for them to degrade naturally; hence, they tend to remain in water bodies for long periods [1–3]. Many dyes are categorized as carcinogenic, which makes them harmful to the environment. Among the various classes of dyes, cationic dyes are a matter of concern, as most of them pose a severe threat to aquatic life, as well as to humans [4,5]. Different types of artificial dyes are present in the effluent released from several process industries, particularly the textile, leather, and paper industries. These industries produce many kinds of dyes worldwide, with approximately 700,000 tons produced per annum [6,7]. The discharge of

colored wastewater into aquatic bodies not only degrades their aesthetic nature but it also hinders the transmission of light from the sun into water bodies [8]. Methylene blue (MB) is a well-known cationic dye that is widely used in the coloration of cotton, silk, and wool, and it may cause health problems such as nausea, vomiting, increased heart rate, tissue necrosis, eye burn, and difficulty in breathing [9,10]. Therefore, it is essential to identify efficient methods to resolve dye pollution issues that are less expensive in terms of operation. To treat the wastewater before releasing it into water bodies, various treatment methods such as photodegradation techniques, activated sludge, and chemical coagulation are applied [11]. However, most dyes are non-oxidizable through traditional wastewater treatment. Nevertheless, dyes can adsorb onto adsorbents; therefore, adsorbents were found to be more efficient and, hence, attracted the attention of many researchers worldwide [12,13].

Recently, the application of nanotechnology flourished due to its immense benefits, attracting researchers to implement nanomaterials for water purification. This approach resulted in the high adsorption removal of pollutants in a cost-effective manner [14,15]. Nanomaterials are appreciated for their high reactivity strength, low mass, and highly active surfaces [16]. Iron nanoparticles were successfully used for the adsorption of dyes such as methylene blue [17], crystal violet [18], phenol red [19], acid red 88 [20], and methyl orange [21]. There are various physicochemical methods used for the synthesis of zero-valent iron (ZVI); however, biogenic and green syntheses using plants attracted the attention of researchers due to their non-toxicity and reusability. Furthermore, plant extracts are able to stabilize ZVI and cap it. Several plant extracts were used for the green synthesis of ZVI and for the treatment of water, such as *Rosa damascene* and *Thymus vulgaris* extract applied for the removal of chromium(VI) [22], tea polyphenols applied for the removal of cationic dyes [23], and black tea extract for the adsorption of a second-generation herbicide ametryn [24].

Hence, the aim of the present research study was to characterize and evaluate the surface adsorptive performance of zero-valent iron nanoparticles synthesized from sweet lime pulp waste (nZVISLP) to adsorb MB from wastewater. The process parameters affecting the process were studied scientifically. The optimal parameters of the isotherm/kinetic models that best represent the adsorption process were estimated using a hybrid evolutionary optimization technique. The novelty of this study is the application of nZVISLP for the removal of MB dye, as well as the application of a differential evolution optimization technique, which is a first in this field.

## 2. Materials and Methods

### 2.1. Adsorbate

MB dye with the chemical formula $C_{16}H_{18}ClN_3S \cdot xH_2O$ was procured from "LOBA Chemie Pvt Ltd., India. Then, 1 g/L MB stock solutions were prepared by diluting the desired amount of MB with distilled water. Various testing samples with initial concentrations of 10–40 mg/L were generated by further dilution of the stock solution.

### 2.2. Synthesis of nZVISLP

To minimize the operational cost of the adsorption process, the adsorbent was prepared from agricultural waste. As a source of low-cost agricultural waste, the pulp of sweet lime waste was obtained from a local juice center. This pulp was initially oven-dried for 24 h, followed by grinding and sieving through 45–100 mesh. Then, 15 g of dried powder of sweet lime pulp was added to 250 mL of pure distilled water. This solution was thoroughly mixed in an orbital shaker at 100 rpm and heated at 80 °C for 1 h. Then, the solution was cooled to room temperature and vacuum-filtered to obtain the extract. Citrus fruits extracts are reported to contain several biomolecules such as flavonoids, alkaloids, antioxidants, and other phenolic compounds [25]. These compounds are also reported to be good reducing agents [26], which reduce metal ions. nZVISLP synthesis was done using the method described by Gautam et al. [19], through the addition of $FeSO_4$ solution (5.8 mol/L) to the sweet lime pulp extract in a ratio of 1:3 (*v/v*). Upon the addition of sweet lime extract to the iron

solution, the solution color changed from brown to black. The pH of the solution was then adjusted by adding NaOH. During this procedure, all metal ions were rapidly transformed into nano-metallic particles due to the addition of NaOH, and the mixture became black. These particles were separated by centrifugation. Then, the black particles were vacuum-dried for 24 h at 50 °C, and stored in a vacuum desiccator before further use for MB adsorption. The various stages of nZVISLP synthesis from sweet lime waste pulp to iron particles are shown in Figure 1.

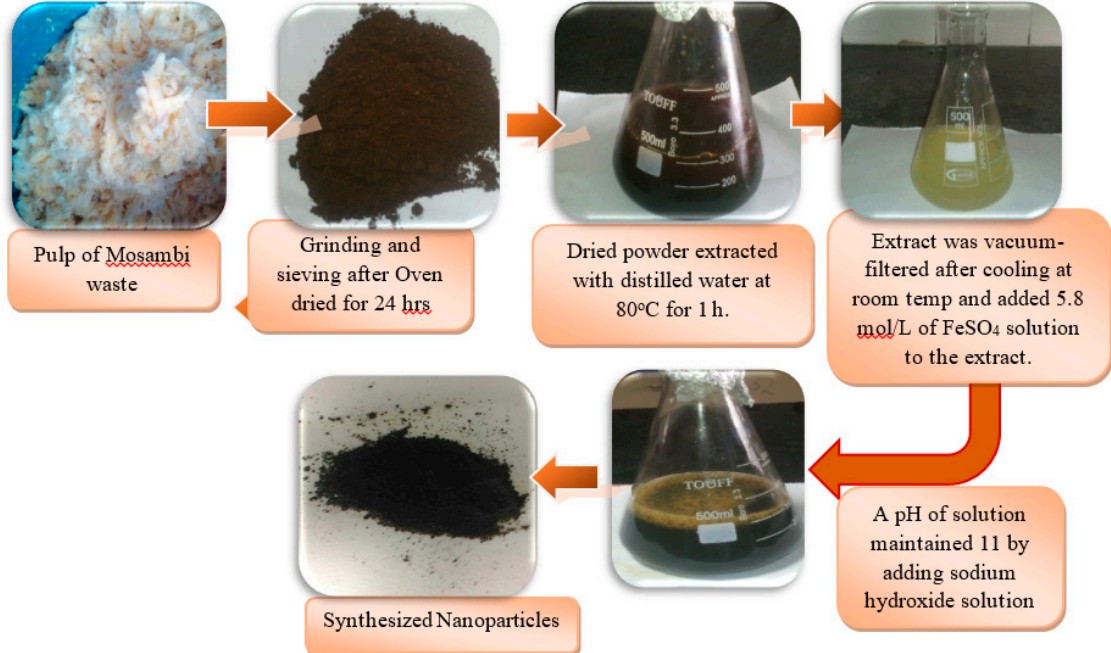

**Figure 1.** Different steps for the synthesis of zero-valent iron nanoparticles synthesized from sweet lime pulp waste (nZVISLP).

### 2.3. Characterization of MB-Adsorbed nZVISLP

Fourier-transform infrared (FT-IR) analysis of the nZVISLP and MB-adsorbed nZVISLP was conducted using infrared spectroscopy (Model Number-Nicolet$^{TM}$6700, Thermo Scientific, Waltham, MA, USA) to evaluate the functional groups in nZVISLP before and after the adsorption process for each sample. FT-IR spectra were scrutinized within the range of 500–6000 cm$^{-1}$. The morphology of the synthesized nZVISLP and the elemental analysis were carried out using SEM (Model Number-JSM-6490LV, Make-JEOL, Tokyo, Japan). For SEM analysis, dried powder of nZVISLP was dispersed in ethanol and sonicated, and then a thin film of sample was prepared on glass and coated with platinum before analysis using SEM. A very small amount of dried nZVISLP was spread over carbon tape and coated with platinum for energy-dispersive X-ray spectroscopy (EDX) analysis. The X-ray diffraction (XRD) analysis of the nZVISLP was carried out using a powder X-ray diffractometer (PW 3040/60 PanAlytical, EA Almelo, the Netherlands), and the scanning of the sample was done in the range of 10° to 80° at a rate of 2 min, with Cu K$\alpha$ ($\lambda$ = 1.5406 Å) radiation.

### 2.4. Adsorption Process

To investigate the underlying mechanisms involved in MB adsorption and to identify the inherent batch adsorption kinetics, experimental studies were conducted. The removal of MB depends on various process parameters such as initial MB concentration, residence time, pH, adsorbent dosage, and process temperature; hence, their individual and interactive effects were studied systematically. The impact of pH on the MB removal was studied within the pH range of 2–10. Various adsorbent dosages (0.4, 0.8, 1.0, and 1.2 g/L) were added to 100 mL of a dye solution with various initial

concentrations (10–40 mg/L) in a 250-mL conical flask. These mixtures were stirred at 100 rpm and 25 °C in an orbital shaker. Samples were collected from each conical flask at different time intervals. The amount of MB removed at each time interval was measured using an ultraviolet–visible light (UV–Vis) spectrophotometer at $\lambda_{max}$ = 664 nm. All adsorption experiments were done in triplicate, and the average values were used for further analysis.

*2.5. Isotherm and Kinetic Model Parameter Evaluation*

2.5.1. Equilibrium Isotherm Models

To understand the inherent mechanisms in the MB adsorption process, the equilibrium data obtained from the experiments were fitted to different isotherm models. The conventional solid–liquid adsorption isotherm models (single component) that were evaluated in this study are presented in Table S1 (Supplementary Materials). This table presents both linear and non-linear (conventional) forms of isotherm models. It is common practice by most researchers to linearize non-linear model expressions, as it is not straightforward to calculate constants in the non-linear models. Even though the linearized model equation adequately fits the equilibrium data, providing higher $R^2$, the predicted values give inferior values, while successive constants are substituted in the non-linear model form. Karri et al. [27,28] in their study concluded that the inherent mechanisms of the adsorption process could be underestimated using the non-linear model expression. Thus, when using the non-linearity of the model and when estimating the parameters (constants) in the non-linear isotherm model, we need an appropriate optimization approach or technique, which estimates the optimal set of parameters that best represent the isotherm model based on an excellent statistical error function. Therefore, in this study, root-mean-squared error (RMSE) was used as the error function, as it is the most frequently used statistical measure. RMSE is an excellent error function for measuring the accuracy and aggregating the magnitude of errors; it is computed as follows:

$$\text{RMSE} = \sqrt{\frac{1}{n-1}\sum_{i=1}^{n}\left(q_{e,pred}^{i} - q_{e,exp}^{i}\right)^2} \tag{1}$$

where $i$ is the number of samples (experimental runs), and $q_{e,exp}$ (mg/g) and $q_{e,pred}$ (mg/g) are the experimental and theoretically predicted equilibrium values, respectively. Another statistically significant metric is Pearson's chi-square ($\chi^2$) metric, which measures the goodness of fit between the model-predicted values and the experimental data. The mathematical expression is expressed as follows:

$$\chi^2 = \sum_{i=1}^{n} \frac{\left(q_{e,exp}^{i} - q_{e,\,pred}^{i}\right)^2}{q_{e,pred}^{i}}. \tag{2}$$

This metric gives a smaller value when the model-predicted values are in close agreement with the experimental values, whereas it produces a higher value if they vary distinctly.

2.5.2. Optimization Technique

Most traditional optimization methods involve a trial-and-error approach, which is not only tedious but also makes the process cumbersome. Therefore, hybrid evolutionary methods can be a suitable replacement for conventional methods. Thus, in this research, differential evolution optimization (DEO), which is a hybrid evolutionary technique, was implemented. The benefits, algorithm, and methodology for implementing DEO are presented in the Supplementary Materials.

2.5.3. Equilibrium Kinetic Models

In this research study, four kinetic models, shown in Table S2 (Supplementary Materials), were evaluated. Most researchers estimated the parameters in kinetic models by linearizing them and

applying traditional least square regression methods. As explained in the previous section, equilibrium values predicted from these models give inferior values. Therefore, to overcome the undervaluation and retain the non-linearity associated with the process, the abovementioned DEO technique was also implemented to evaluate the intrinsic parameters; thus, the most suitable kinetic model could be identified.

## 3. Results and Discussion

### 3.1. Characterization of nZVISLP

#### 3.1.1. FT-IR

The FT-IR spectra of synthesized nZVISLP and MB-loaded nZVISLP are presented in Figure 2. Figure 2 reveals many absorption peaks, suggesting the complex structure of the nanomaterial. The broad peak at 3431.3 cm$^{-1}$ was due to the stretching vibration of the bond for the existence of –OH functional groups on the nZVISLP surface. The sharp adsorption peaks at 2855.6 and 2917.7 cm$^{-1}$ were attributed to the symmetrical vibration of –CH$_2$ and asymmetrical vibration of –CH$_3$, respectively. The symmetric and asymmetric bending vibrations of –CH$_3$ and –CH$_2$ were further confirmed by the presence of peaks at 1417.2 and 1337 cm$^{-1}$, respectively. The influential band at 1642.7 cm$^{-1}$ was due to the vibration of –C=O from a carboxylic acid. The peak observed at 1548.9 cm$^{-1}$ was due to the existence of aromatic rings. The sharp tiny peaks observed at 1012.5 cm$^{-1}$ were attributed to the bending vibration of –OH and stretching vibration of –C–O–C– in the lignin structure of the material. As presented in Figure 2, for the MB-loaded nZVISLP, the peak of the stretching vibration of –OH groups at 3431.3 cm$^{-1}$ shifted to 3419 cm$^{-1}$, whereas the peak representing the –C=O stretching vibration of carboxylic acid with intermolecular hydrogen bonds also shifted from 1642.7 to 1646 cm$^{-1}$. The symmetric and asymmetric vibration peaks of –CH$_2$ and –CH$_3$ also shifted to higher wavelengths after MB loading. These results indicate that the active sites including –OH and –C=O groups existing on the surface of the material may lead to the potential removal of a cationic dye, such as MB [29], along with iron particles.

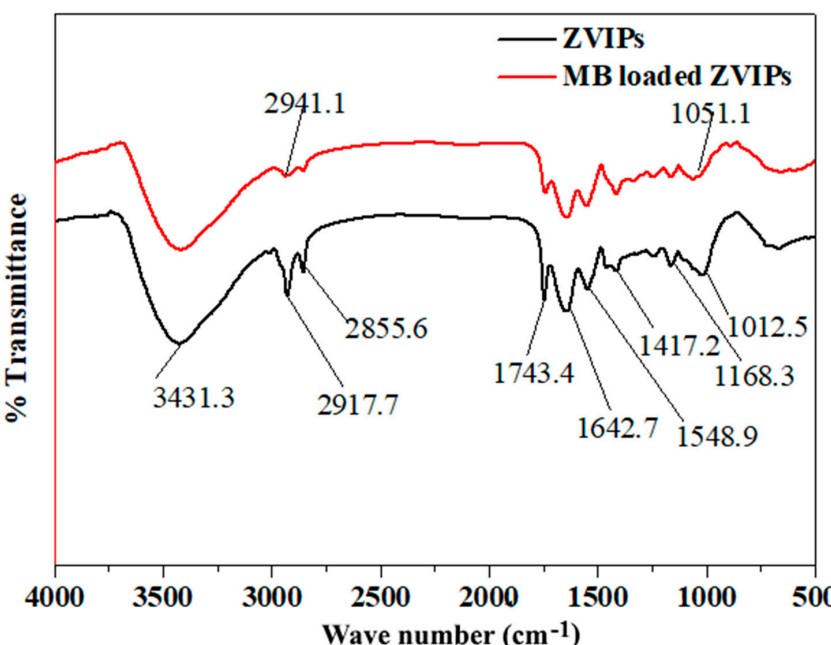

**Figure 2.** Infrared spectra of nZVISLP before and after adsorption of methylene blue (MB).

### 3.1.2. SEM and EDX

SEM analysis revealed the morphological characteristics and surface features of nZVISLP. As shown in Figure 3a,b, before adsorption, the surface of the material possessed a rough structure and exhibited a different micromorphology. It was also observed that nZVISLP exhibited different sizes in the range of 86–113 nm, and the particles were clumped in aggregates. EDX analysis (Figure 3c and Table 1) indicated the presence of Fe, C, $O_2$, and Ca in nZVISLP. C and O could have been due to the carbonaceous and oxygenated phytochemicals of sweet lime pulp extract. The small $O_2$ peak observed in Figure 3c may also signify the occurrence of oxidized Fe particles [30].

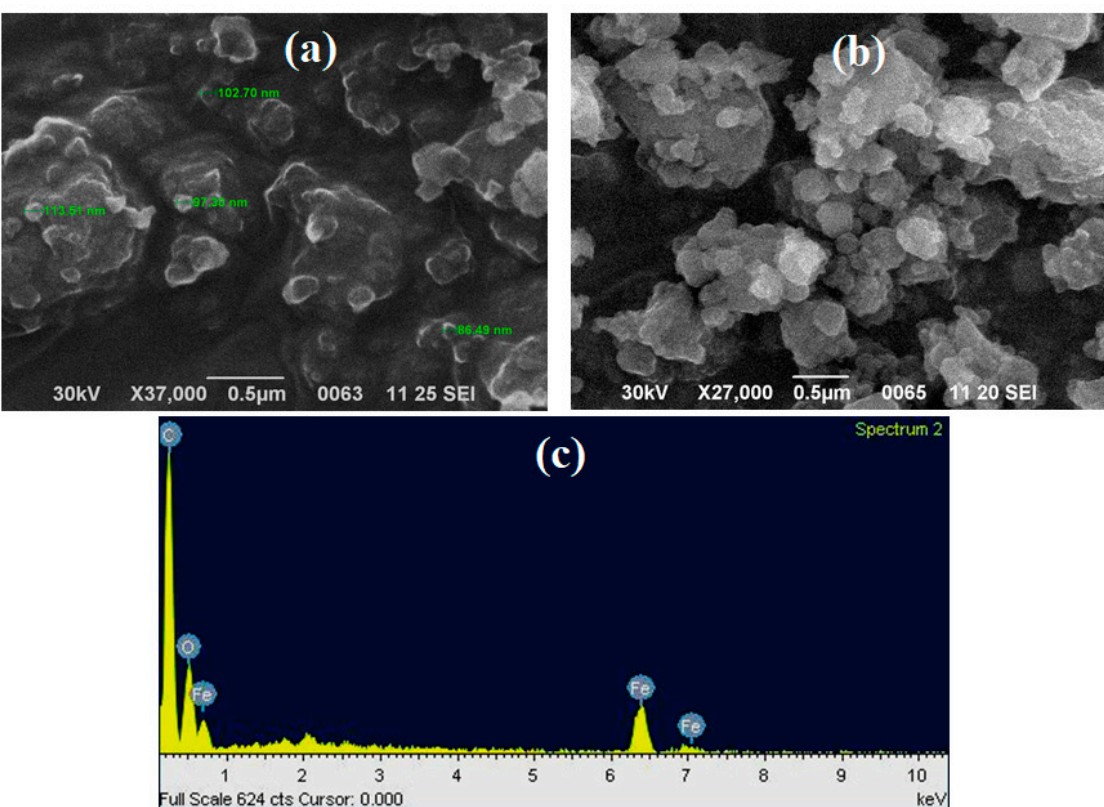

**Figure 3.** (**a,b**) SEM images of nZVISLP; (**c**) SEM energy-dispersive X-ray spectroscopy (EDX) analysis.

**Table 1.** Elemental composition of zero-valent iron nanoparticles synthesized from sweet lime pulp waste (nZVISLP).

| Element | Weight (%) |
|---------|------------|
| C (K) | 54.31 |
| O (K) | 15.03 |
| Si (K) | 1.71 |
| Ca (K) | 3.10 |
| Fe (K) | 11.98 |
| Zr (L) | 13.86 |
| Total | 100.00 |

### 3.1.3. Zero-Point Charge

The zero-point charge ($pH_{ZPC}$) is an essential characteristic of any adsorbent, and it represents the particular pH value at which the surface charge of that adsorbent is neutral. The determination of $pH_{ZPC}$ of the synthesized adsorbent was done using a 0.01 M solution of NaCl, the pH of which was adjusted in the range of 2 to 10 by adding 0.1 M HCl and 0.1 M NaOH. For this, 0.20 g of the prepared

nZVISLP was added to 50 mL of NaCl (0.01 M) in a conical flask, and then the solution pH was adjusted in the range of 2 to 10, stirring for 48 h [31]. Then, the final pH of the solution was determined. The point at which both pH curves crossed was specified as the pH$_{ZPC}$ of the adsorbent. The pH$_{ZPC}$ of the synthesized nZVISLP was determined at different pH (2–10) (see Figure S1, Supplementary Materials). The pH$_{ZPC}$ of nZVISLP was found to be 4.5, representing that the surface charge of the material is neutral at this pH, whereas it is positive below this pH.

### 3.1.4. Particle Size and Zeta Potential Analysis

The particle size of the sweet lime synthesized adsorbent was determined using a Zeta nanosizer (Nano-ZS90, Malvern, UK), after homogenization in $C_2H_5OH$ at room temperature. The results obtained are presented in Figure 4a. This profile indicates a broad peak between 200 and 500 nm. The zeta potential of nZVISLP was also determined using the same equipment. The result of the zeta potential analysis is shown in Figure 4b. The magnitude of the zeta potential defines the stability of the adsorbent in a dispersion medium. It signifies the electrostatic repulsion between the adsorbent and the dispersion medium [32]. The result indicates that the magnitude of zeta potential increased at higher pH, depicting that the synthesized adsorbent was stable at higher pH.

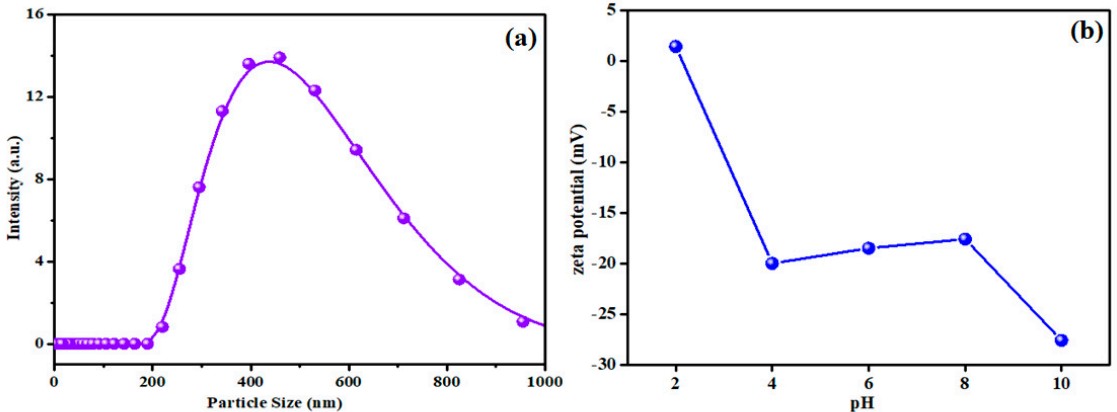

**Figure 4.** (**a**) Particle size and (**b**) zeta potential of nZVISLP.

### 3.1.5. Determination of Porosity and Internal Structure

nZVISLP was analyzed by determining its Brunauer–Emmett–Teller (BET) surface area at 77 K for the determination of its porosity and internal structure (see Figure S2, Supplementary Materials). The adsorption–desorption of MB (Figure S2a, Supplementary Materials) showed an increase in the adsorption of condensed nitrogen onto the pores with an increase in pressure, and vice versa for desorption at the same temperature. Figure S2b (Supplementary Materials) shows the variation of pore radius ($r_p$) with respect to pore volume/pore radius ($dV_p/dr_p$). It can be observed that $dV_p/dr_p$ decreased with an increase in pore radius. The values of pore volume, mean pore radius, and surface area of the synthesized adsorbent were 0.061 cm$^3$/g, 12.20 Å, and 40.87 m$^2$/g, respectively. Konicki et al. [33] also prepared an iron graphite nano-composite with a surface area of 47 m$^2$/g for the removal of methylene blue. Figure S2c (Supplementary Materials) shows the BET plot where p/V (p − p$_0$) against p/p$_0$, which shows a linear relationship. The surface area of nZVISLP was also calculated using the size determined from dynamic light scattering (DLS), which was very small (0.2 m$^2$/g), and that determined from BET, which was 47 m$^2$/g. The high difference between both values represents the porous nature of the adsorbent, which enhanced the adsorption of N$_2$ in the BET analysis.

### 3.1.6. X-Ray Diffraction (XRD)

The synthesized adsorbent was analyzed through XRD to determine its crystallinity, and the pattern is shown in Figure S3 (Supplementary Materials). The absence of any distinct peak confirmed

that the green-synthesized iron nanoparticles were amorphous in nature. This result indicates the amorphous nature of nZVI, and the XRD results were very similar to experimental studies reported by Machado et al. [34] and Wang et al. [15], who also reported the amorphous nature of iron nanoparticles synthesized through a green route. The peak at 2θ 34.8° was attributed to iron oxide, and the peaks at 2θ 45.3° and 64.3° represented the zero-valent iron; these values are in close agreement with the reported XRD spectrum for zero-valent iron by Krzisnik et al. [35].

### 3.2. Adsorption Study

The influence of process parameters (equilibrium time, pH, initial concentration of MB, adsorbent dosage, and process temperature) on MB removal using nZVISLP was investigated systematically. In this regard, the experiments were conducted with each variable in different ranges as follow: $C_0$ (5–40 mg/L), contact time (0–300 min), pH (4–10), temperature (25–45 °C), and adsorbent dosage (0.4–1.2 g/L). The analyses of these experiments are described below.

3.2.1. Impact of MB Initial Concentration ($C_0$) and Equilibrium Time

The impact of $C_0$ (varied from 5–40 mg/L) on percentage removal was investigated at 25 °C, with a pH of 8 and an nZVISLP dosage of 1.2 g/L. The effect of different $C_0$ values of MB uptake obeyed an inverse exponential form (see Figure 5a), i.e., rapid in the initial stages and steady in the later stages. This signifies the possibility of monolayer adsorption taking place on the surface of nZVISLP. The increasing trend in MB removal continued until it reached an equilibrium state (~180 min). This may have been due to the fact that the vacant sites on the surface of nZVISLP were only available in the early stages of the adsorption process. Typically, 80% of the adsorbate (with $C_0$ = 10 mg/L) was removed within 30 min of the adsorption process (see Figure 5a), and the process then proceeded at a slow pace, reaching a maximum removal of 91% in 300 min. However, an increase in $C_0$ resulted in a decrease in removal concentration. Upon increasing dye concentration, the adsorption site became saturated, which limited the removal of MB [36]. As a result, $q_t$ increased from 8.20 ± 0.01 to 32.6 ± 0.02 mg/g with a rise in $C_0$ from 10 mg/L to 40 mg/L. The decrease in $q_t$ with the increase in $C_0$ potentially occurred as a result of the initial concentration providing the driving force to overcome the mass transfer resistance [37].

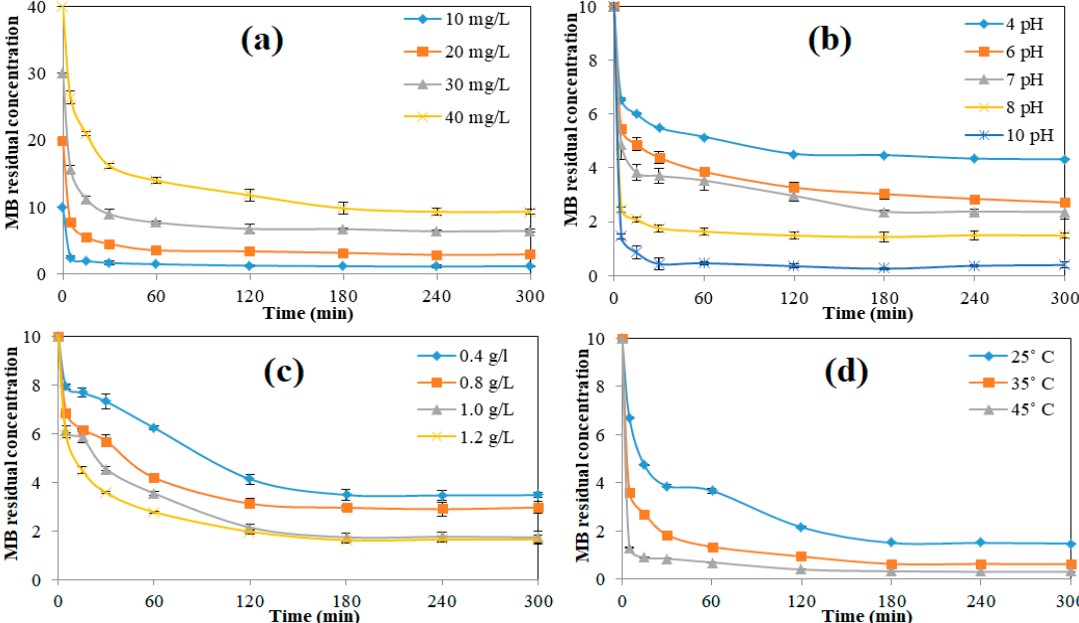

**Figure 5.** Influence of various parameters on MB removal: (**a**) $C_0$; (**b**) pH; (**c**) adsorbent dosage; (**d**) process temperature.

### 3.2.2. Impact of Solution pH

The impact of pH on MB adsorption by nZVISLP was investigated by varying the solution pH within the range of 4–10 for a dosage of 1.2 g/L and $C_0$ of 10 mg/L, conducted at 25 °C. A 0.1 N HCl/NaOH mixture was added to maintain the pH at the required value. As seen in Figure 5b, the MB removal (%) was enhanced with a rise in solution pH. The pH of the solution was also checked after the adsorption process at different time intervals, whereby it was found that the solution pH changed slowly with time within the initial 60 min and then became constant, resulting in the adsorption changing in the initial 60 min and then becoming almost constant (Figure 5b). The pH of the solution changed to approximately its $pH_{ZPC}$ value. At acidic pH, a lower removal was observed due to the competitive interactions of $H^+$ and catatonic MB, as well as the repulsive force interaction between the dye cations and the positively charged sites. This was confirmed by the stable zeta potential magnitude between pH 4.0 and 8.0 (Figure 4b). Upon increasing pH (8.0 to 10.0), the MB adsorptive removal increased, due to the increase in the number of negatively charged sites, which led to the existence of an electrostatic force of attraction between the cationic MB and nZVISLP [38]. These results also confirmed the results reported in earlier studies [38–40], which showed an increase in cationic dye adsorption with a rise in pH. MB adsorption was almost constant at alkaline pH values (8–10).

### 3.2.3. Impact of nZVISLP Dosage

The effect of nZVISLP dosage on MB adsorption was also systematically investigated by varying the amount of nZVISLP from 0.4 to 1.2 g/L, keeping other process parameters at fixed values (pH: 8, $C_0$: 10 mg/L, and temperature: 25 °C). Figure 5c shows that the adsorptive removal of MB increased from 71–86% with the increase in nZVISLP dosage from 0.4 to 1.0 g/L, as evidenced by the $C_0$ value changing from 3.4 ± 0.2 to 1.6 ± 0.1 mg/L,. With a further increase in dosage (up to 1.2 g/L), there was no substantial increase observed in MB removal. Upon increasing nZVISLP dosage from 0.4 to 1.2 g/L, $q_{max}$ values increased from 6.5 ± 0.2 to 8.2 ± 0.01 mg/g. Upon increasing adsorbent dosage, the surface area available for adsorption increases, which then enhances the adsorption [41]. Li et al. [39] also reported a similar trend for MB adsorption. However, a further increase in dosage did not influence the removal rate, due to the unavailability of adsorption sites [38].

### 3.2.4. Impact of Process Temperature

The impact of temperature on MB removal efficiency was also investigated at different temperatures (25, 35, and 45 °C). As shown in Figure 5d, the removal of MB increased from 86.4 ± 0.05% to 97.0 ± 0.05% as the temperature was increased from 25 to 45 °C. On the other hand, a state of equilibrium was reached within 60 min at 45 °C as compared to the adsorption process conducted at 25 °C (240 min). In this research study, maximum MB removal was obtained at the highest temperature of 45 °C, which was consistent with the result of previously reported studies [42]. The diffusion rate of MB molecules through the internal pores of nZVISLP and the external boundary layer may be slightly increased by increasing the temperature.

### *3.3. Identification of Equilibrium Isotherm Models*

The removal of MB from the aqueous solution may depend on various mechanisms such as reduction or oxidation, or degradation in the presence of OH· radicals. In the presence of a reducing agent, MB is reduced to leuco methylene blue (LMB), which is a colorless compound; upon oxidation by agitation, LMB is transformed back to its colored state, MB [43]. However, in this study, after removal of the particles, the colorless MB solution was not re-colored upon constant agitation, which confirms no participation of reduction in MB removal. The removal of MB was mainly governed by adsorption.

To get insight into the adsorption mechanisms of nZVISLP for MB removal and to verify the performance of the adsorbent, various isotherms, as indicated in Table S1 (Supplementary Materials), were validated. The isotherm parameters were evaluated using DEO with an objective function

to minimize statistical metrics including $\chi^2$ and RMSE. The performance of the optimal isotherm parameters is presented in Figure 6. These results indicate that linearizing the non-linear equation underestimated the features of the actual isotherm. It can be observed that the values predicted by DEO using optimized parameters were pretty close to the experimental values. To further confirm the performance and verify the statistical significance, the previously discussed prominent statistical metrics were evaluated and compared for both linear and non-linear model forms, as shown in Table 2. Again, these results further indicated that the parameters estimated using the non-linear model provided a higher $R^2$. For all the evaluated isotherm models, $R^2$ was higher than 0.94, and the $R^2$ values for the Freundlich and Sips isotherm models were close to 0.98, thus validating their better description of nZVISLP adsorption characteristics for MB removal. Another essential statistical metric, $\chi^2$, indicated that the values were ≥0.98, thus confirming the suitability of these isotherms. All other significant statistical measurements that were evaluated showed a similar trend. As a result, the optimal isotherm parameters estimated by implementing DEO better captured the adsorption process of MB onto the surface of nZVISLP.

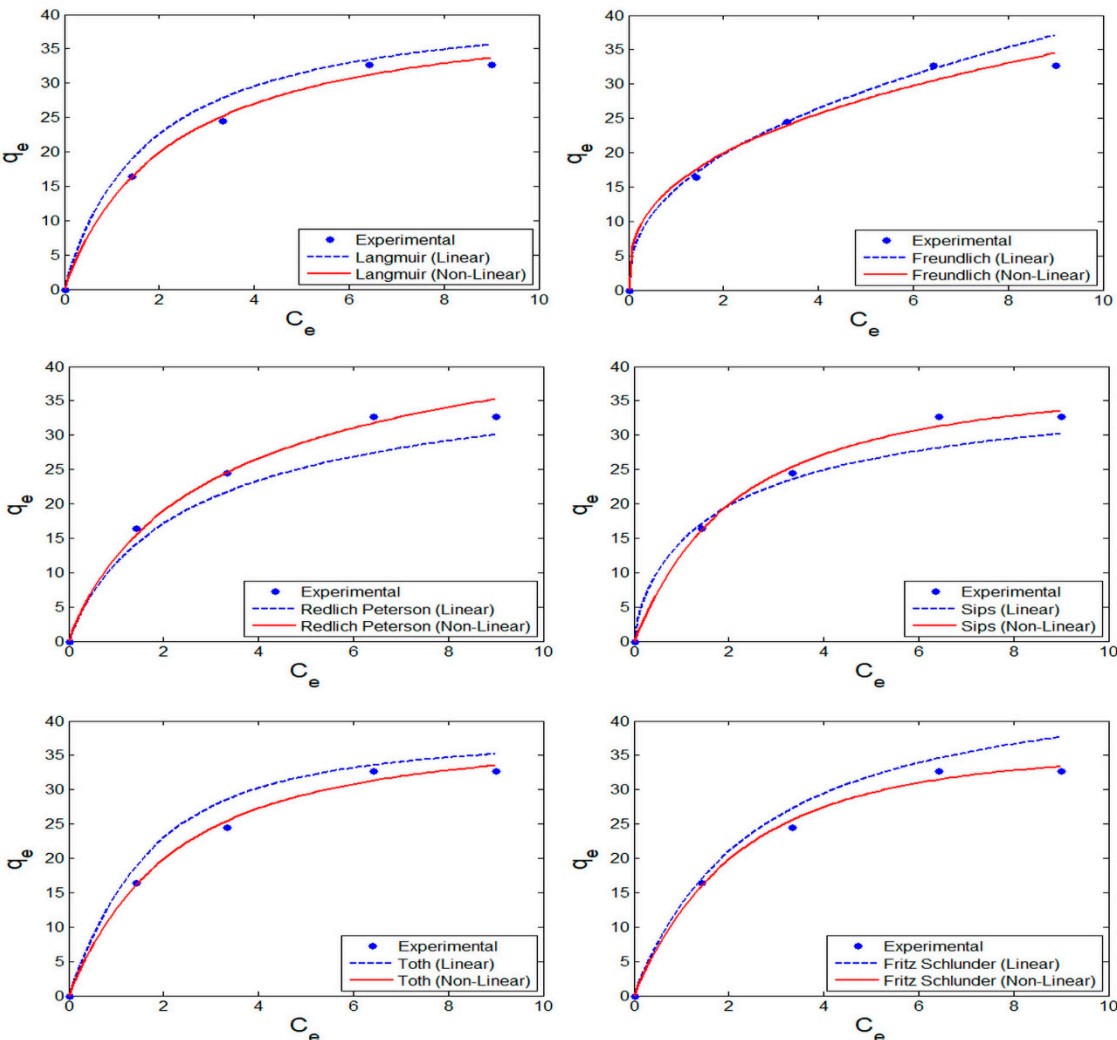

**Figure 6.** Scatter plots depicting the performance of differential evolution optimization (DEO) in terms of linear transformation for various isotherm models.

**Table 2.** Comparison of prominent statistical metrics using linear (L) and non-linear (NL) approaches for evaluating the optimal isotherm model parameters.

| | Models → | Freundlich | | Langmuir | | Redlich–Peterson | | Sips | | Toth | | Fritz–Schlunder | |
|---|---|---|---|---|---|---|---|---|---|---|---|---|---|
| Metrics ↓ | Expressions ↓ | L | NL | L | NL | L | NL | L | NL | L | NL | L | NL |
| | Parameters → | $K_L$: 42.68 $b_F$: 0.563 | $K_L$: 41.99 $b_F$: 0.450 | $K_L$: 14.76 $b_L$: 2.38 | $K_L$: 15.44 $b_L$: 2.735 | $K_R$: 17.93 $a_R$: 0.572 $\alpha$: 0.926 | $K_R$: 18.60 $a_R$: 0.511 $\alpha$: 0.907 | $K_s$: 40.210 $b_s$: 0.569 $n_s$: 1.156 | $K_s$: 40.064 $b_s$: 0.462 $n_s$: 0.911 | $K_{th}$: 38.21 $b_{th}$: 0.353 $n_{th}$: 1.423 | $K_{th}$: 38.97 $b_{th}$: 0.336 $n_{th}$: 1.222 | $K_{FS}$: 51.04 $b_{FS}$: 0.354 $n_{FS}$: 0.982 | $K_{FS}$: 59.20 $b_{FS}$: 0.265 $n_{FS}$: 0.878 |
| Correlation coefficient of determination ($R^2$) | $\dfrac{\sum_{i=1}^{n}\left(q_{e,pred}^{i}-q_{e,exp}^{i}\right)^2}{\sum_{i=1}^{n}\left[\left(q_{e,pred}^{i}-q_{e,mean,exp}^{i}\right)^2\right]}$ | 0.979 | 0.979 | 0.940 | 0.945 | 0.973 | 0.974 | 0.973 | 0.979 | 0.971 | 0.975 | 0.971 | 0.973 |
| Average relative error (ARE) | $\dfrac{100}{p}\sum_{i=1}^{n}\dfrac{\left|q_{e,pred}^{i}-q_{e,exp}^{i}\right|}{q_{e,exp}^{i}}$ | 6.378 | 1.742 | 3.153 | 3.470 | 7.880 | 2.672 | 3.098 | 1.952 | 6.660 | 2.005 | 5.711 | 2.101 |
| Sum of the squares of errors (SSE) | $\sum_{i=1}^{n}\left(q_{e,pred}^{i}-q_{e,exp}^{i}\right)^2$ | 26.429 | 3.900 | 20.814 | 10.107 | 48.309 | 8.547 | 11.106 | 3.790 | 29.314 | 3.716 | 36.175 | 3.459 |
| Sum of the absolute errors (SAE) | $\sum_{i=1}^{n}\left|q_{e,pred}^{i}-q_{e,exp}^{i}\right|^2$ | 9.439 | 3.285 | 5.754 | 5.789 | 12.994 | 4.536 | 5.199 | 3.466 | 9.825 | 3.494 | 10.170 | 3.484 |
| Root-mean-square error (RMSE) | $\sqrt{\dfrac{1}{n-1}\sum_{i=1}^{n}\left(q_{e,pred}^{i}-q_{e,exp}^{i}\right)^2}$ | 2.099 | 0.806 | 1.863 | 1.298 | 2.838 | 1.194 | 1.360 | 0.795 | 2.210 | 0.787 | 2.455 | 0.759 |
| Pearson's chi-square measure | $\chi^2=\sum_{i=1}^{n}\dfrac{\left(q_{e,exp}^{i}-q_{e,\,pred}^{i}\right)^2}{q_{e,pred}^{i}}$ | 0.905 | 0.989 | 0.902 | 0.950 | 0.877 | 0.964 | 0.940 | 0.988 | 0.779 | 0.989 | 0.788 | 0.989 |
| Hybrid fractional error function (HYBRID) | $\dfrac{100}{n-p}\sum_{i=1}^{n}\dfrac{\left(q_{e,pred}^{i}-q_{e,exp}^{i}\right)^2}{q_{e,exp}^{i}}$ | 17.785 | 1.977 | 10.407 | 5.611 | 27.241 | 4.599 | 5.980 | 1.999 | 19.943 | 1.995 | 19.100 | 1.973 |

$q_{e,exp}^{i}$, $q_{e,pred}^{i}$ are the $q_e$ values of the experimental and predicted values, respectively; $p$ is the number of parameters; $n$ is the number of experimental runs. Units: Langmuir: $K_L$ (mg/g), $b_L$ (L/mg); Freundlich: $K_F$ (mg$^{-1-(1/n)}$ L$^{(1/n)}$ g$^{-1}$), $b_F$ is dimensionless; Temkin: $K_T$ (L/mg), $b_T$ (J/mol); Redlich–Peterson: $K_R$ (L/g), $a_R$(L/mg) $^{nP}$, $n_R$ is dimensionless; Sips: $K_S$ (mg/g), $b_s$ (L/mg)$^n$, $n_s$ is dimensionless; Toth: $K_{th}$ (L/g), $b_{th}$ (mg/L)$^n$, $n_{th}$ is dimensionless. Fritz–Schlunder: $K_{FS}$ (mg/g), $b_{FS}$ (mg/L), $n_{FS}$ is dimensionless.

### 3.4. Evaluation of Intrinsic Parameters in Kinetic Models

The parameters of the four kinetic models were assessed using their non-linear forms (see Table S3, Supplementary Materials), and the DEO method was implemented for the models. The efficacy of the optimal characteristic parameters of the four kinetic models is presented in Figure 7. Again, it can be observed that linearization undervalued the kinetic parameters. For all the kinetic models, the DEO using optimized kinetic parameters better predicted the experimentally obtained values. These scatter plots validated the non-linearity of the adsorption process; hence, the DEO-based non-linear model predictions better correlated with the experimental values. To further confirm the model predictions, non-linear statistical metrics were evaluated. The prominent statistical metrics were evaluated and compared for both linear and non-linear kinetic model expressions, as shown in Table 3 for $C_0$ of 10 mg/L. The linear plots of the pseudo first-order (PFO), pseudo second-order (PSO), and Weber–Morris kinetic models $C_0$ of 20, 30, and 40 mg/L are shown in Figures S4 and S5 (Supplementary Materials). For all the evaluated kinetic models, $R^2$ was higher for non-linear models, thus confirming that these kinetic models better describe the MB adsorption onto nZVISLP. The $R^2$ values were ≥0.98 for the pseudo second-order kinetic model, thus confirming that this model better explains the MB adsorption process. Also, all other prominent statistical metrics resulted in a similar trend. It can be observed from Table 3 that the optimal parameters derived from DEO not only provided lower values of RMSE and $\chi^2$, but also provided improved values for other statistical metrics. Therefore, the optimal kinetic parameters estimated by DEO suitably captured the dynamics of MB adsorption onto the surface of nZVISLP.

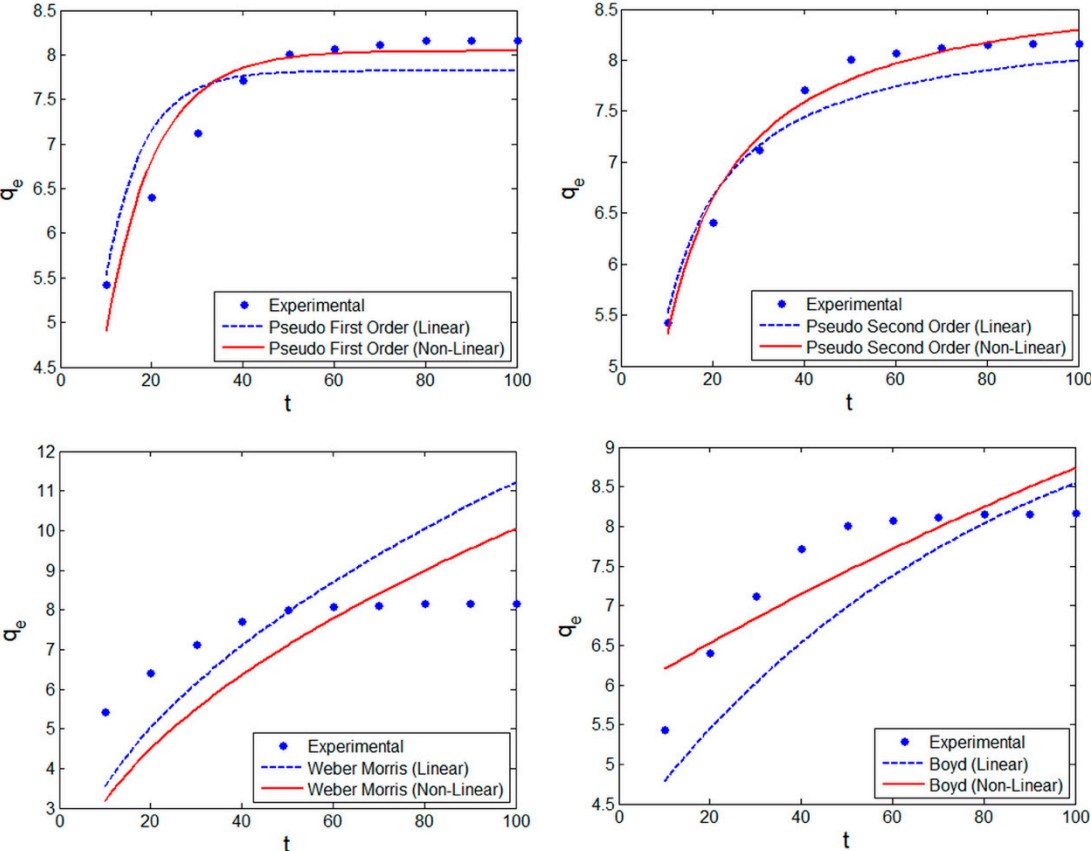

**Figure 7.** Scatter plots comparing the performance of DEO in terms of linear transformation for various kinetic models.

**Table 3.** Comparison of prominent statistical metrics using linear (L) and non-linear (NL) approaches for evaluating the optimal kinetic model parameters.

| Models → | | PFO | | PSO | | Weber–Morris | | Boyd | |
|---|---|---|---|---|---|---|---|---|---|
| **Metrics ↓** | **Expressions ↓** | **L** | **NL** | **L** | **NL** | **L** | **NL** | **L** | **NL** |
| | Parameters → | $K_1$: 0.0938 $q_e$: 8.0228 | $K_1$: 0.1226 $q_e$: 7.818 | $K_2$: 0.0170 $q_e$: 8.8485 | $K_2$: 0.0226 $q_e$: 8.418 | $K_{id}$: 1.123 | $K_{id}$: 1.005 | B: 0.0038 $q_e$: 14.9468 | B: 0.0126 $q_e$: 10.326 |
| Correlation coefficient of determination ($R^2$) | $\dfrac{\sum_{i=1}^{n}\left(q_{e,pred}^{i}-q_{e,exp}^{i}\right)^2}{\sum_{i=1}^{n}\left[\left(q_{e,pred}^{i}-q_{e,mean,exp}^{i}\right)^2\right]}$ | 0.9219 | 0.9514 | 0.9362 | 0.9778 | 0.8378 | 0.8862 | 0.7430 | 0.7825 |
| Average relative error (ARE) | $\dfrac{100}{p}\sum_{i=1}^{n}\dfrac{\left|q_{e,pred}^{i}-q_{e,exp}^{i}\right|}{q_{e,exp}^{i}}$ | 3.879 | 2.790 | 2.718 | 1.444 | 17.388 | 16.206 | 8.247 | 4.741 |
| Sum of the squares of errors (SSE) | $\sum_{i=1}^{n}\left(q_{e,pred}^{i}-q_{e,exp}^{i}\right)^2$ | 3.185 | 2.077 | 2.316 | 1.158 | 14.252 | 12.705 | 6.637 | 3.792 |
| Sum of the absolute errors (SAE) | $\sum_{i=1}^{n}\left|q_{e,pred}^{i}-q_{e,exp}^{i}\right|^2$ | 1.363 | 0.718 | 0.640 | 0.174 | 27.877 | 20.285 | 5.802 | 1.930 |
| Root-mean-squared error (RMSE) | $\sqrt{\dfrac{1}{n-1}\sum_{i=1}^{n}\left(q_{e,pred}^{i}-q_{e,exp}^{i}\right)^2}$ | 0.389 | 0.283 | 0.267 | 0.139 | 1.760 | 1.501 | 0.803 | 0.463 |
| Pearson's chi-square measure ($\chi^2$) | $\sum_{i=1}^{n}\dfrac{\left(q_{e,exp}^{i}-q_{e,\ pred}^{i}\right)^2}{q_{e,pred}^{i}}$ | 0.967 | 0.984 | 0.987 | 0.991 | 0.937 | 0.971 | 0.958 | 0.982 |
| Hybrid fractional error function (HYBRID) | $\dfrac{100}{n-p}\sum_{i=1}^{n}\dfrac{\left(q_{e,pred}^{i}-q_{e,exp}^{i}\right)^2}{q_{e,exp}^{i}}$ | 2.142 | 1.290 | 0.925 | 0.274 | 41.831 | 33.549 | 9.019 | 3.127 |

$q_{e,exp}^{i}$, $q_{e,pred}^{i}$ are the $q_e$ values of the experimental and predicted values, respectively; $p$ is the number of parameters; $n$ is the number of experimental runs. PFO—pseudo first-order model; PSO—pseudo second-order model.

*3.5. Thermodynamic Studies*

Thermodynamic studies can present insight into the nature of the adsorption process. These are evaluated from the concepts of enthalpy ($\Delta H°$), Gibbs free energy ($\Delta G°$), and entropy ($\Delta S°$). The procedure for calculating these parameters is given in Table S4 (Supplementary Materials). The values of $\Delta G°$ (kJ/mol) were found to be −20.5, −21.1, and −21.8, at 25, 35, and 45 °C, respectively (from the plot of ln $K_c$ versus 1/T, shown in Figure S6, Supplementary Materials), which confirmed the spontaneity of the adsorption process. The values of $\Delta H°$ (J/mol) and $\Delta S°$ (J/mol K) were found to be 18.7 and 68.67, respectively. The positive values of $\Delta H°$ reconfirmed that the adsorptive removal of MB onto nZVISLP is an endothermic adsorption process. A positive value of $\Delta S°$ indicates the presence of high randomness at the interface of adsorbent/adsorbate while at the adsorption equilibrium.

## 4. Conclusions

The nano-bioadsorbent prepared from sweet lime pulp waste (nZVISLP) showed exciting results in MB removal from an aqueous solution. This nano-bioadsorbent was most effective at alkaline pH (10), whereas the acidic conditions were not favorable. The equilibrium values predicted using the DEO-based optimized isotherm parameters were in close agreement with the experimental values. The $R^2$ values were >0.95 for all studied isotherms, and the $R^2$ values for the Freundlich and Sips isotherm models were ~0.98, thus validating that these isotherms best depict MB adsorption onto nZVISLP. The higher $\chi^2$ values (≥0.98) also confirmed the suitability of these isotherms. Similarly, the kinetic model parameters evaluated by DEO were able to capture the kinetics of MB adsorption, thus resulting in higher $R^2$ and lower RMSE. The endothermic and spontaneous nature of the adsorptive removal of MB onto nZVISLP was confirmed by the results of the thermodynamic study. The treatment of wastewater and the application of prepared nZVISLP may reduce pollution loads in surface water. The prepared iron particles are highly active for the adsorptive treatment of MB-contaminated water.

**Supplementary Materials:** The following are available online at http://www.mdpi.com/2076-3417/9/23/5112/s1: Table S1. Various forms of isotherm models for single component expressed in linear and non-linear forms. Table S2. Summary of different conventional type of kinetics both in linear and non-linear forms generally used to estimate the adsorption rate along with corresponding plot to estimate the parameters. Table S3. Pseudo first order and second order kinetics for the removal of MB by nZVISLP. Table S4: Thermodynamic parameters of MB adsorption onto the nZVISLP. Figure S1. pHzpc of synthesized SLP-IPs. Figure S2. (a) Pressure variation on adsorption/desorption of N2 (b) variation in dVp/drp (c) BET plot. Figure S3. XRD pattern of prepared nZVISLP to identify crystallinity of particles and confirmation of zero valent iron. Figure S4. The pseudo 1st order and pseudo 2nd order kinetics study for the adsorption of MB onto nZVISLP at different initial dye concentrations (C0 = 10, 20, 30 and 40 mg/L; T = 293 K; adsorbent concentration=1.2g/L; initial pH 8.0). Figure S5. Intraparticle diffusion model for the adsorption of MB on to nZVISLP with different initial dye concentrations (C0 = 10, 20, 30 and 40 mg/L; T = 293 K; adsorbent concentration=1.2g/L; initial pH 8.0). Figure S6: Plots of lnKC versus 1/T of the adsorption of MB onto the nZVISLP.

**Author Contributions:** Conceptualization, J.R.K. and J.S.; methodology, J.S., N.S., and S.R.; software, R.R.K.; validation, R.R.K., J.R.K., and J.S.; formal analysis and investigation, N.S., J.-S.C., S.L., and S.R.; data curation, N.S. and J.-S.C.; writing—original draft preparation, N.S. and S.R.; writing—review and editing, J.R.K., J.S., and R.R.K.; visualization, J.-S.C. and S.L.; supervision, J.S.; project administration, J.S.

**Funding:** This research was funded by the University Grant Commission (Grant No. F. 30-382/2017, BSR (UGC-BSR Research start-up grant), and the Science and Engineering Research Board (SERB), Department of Science and Technology (Grant No. ECR/2016/001924), Government of India. In addition, financial support was obtained from a Kwangwoon University Research Grant 2019, Seoul, Korea.

**Acknowledgments:** The authors are grateful to the University Grant Commission (Grant No. F. 30-382/2017, BSR (UGC-BSR Research startup grant), and the Science and Engineering Research Board (SERB), Department of Science and Technology (Grant No. ECR/2016/001924), Government of India for the financial support of this research, as well as the Kwangwoon University Research Grant 2019, Seoul, Korea.

**Conflicts of Interest:** The authors declare no conflicts of interest.

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
