# Peer review of "Process Optimization and Modeling of Methylene Blue Adsorption Using Zero-Valent Iron Nanoparticles Synthesized from Sweet Lime Pulp"

_applsci, doi:10.3390/app9235112_

Round 1
Reviewer 1 Report
Attached

Author Response
We appreciate the valuable comments of all the reviewers on our manuscript and express our sincere thanks for their time and constructive suggestions. The followings are the justifications presented in reply to reviewers’ comment. The critical comments and useful suggestions have been helped us to improve our paper considerably. As indicated in the reply’s that follows, we have taken these comments and suggestions into account, and amended/corrected the statements in the revised version of our manuscript which are with text with yellow background color in the revised manuscript.
Response to Reviewer 1:
This manuscript presented the synthesis, characterization, and reactivity of nZVISLP as a low-cost material to remove dyes from aqueous solution. Methylene blue was used as a model compound. The topic of the manuscript should be of interest to the readers of Applied Science. However, the manuscript is lack of appropriate literature review relevant to MB removal by nZVI or sorbing materials in the introduction and did not clearly state why nZVISLP is worth to be studied in comparison to previous works. I am also concerned the authors overstated or misinterpreted some results (e.g. SEM, sorption as a dominant removal mechanism) based on speculation or without reasonable assumption/discussion. Additionally, some sections were not well organized to carry out the content.
The result section contains some method materials and I suggest to organize characterization results concisely rather listing the results. I believe that this manuscript can be improved with rewritten introduction, additional information on method and reorganization of the contents and results with tables and results. Thus, I recommend it to be reconsidered with major revision for publication. Specific comments are below:
Response to the comment/suggestion:
The manuscript has been modified based on the three reviewers comments, thus the whole manuscript is overhauled.
Comment #1: L39. It is overstated. Dyes are generally biodegradable even though the biodegradation rates are slow.
Response to the comment: As suggested, this sentence has been modified. Page no. 1 line no. 38-39 of revised manuscript.
Comment #2: L57-61. The statement is not rational since nanomaterials and AC are not the same category.
Response to the comment: Sorry for the confusing statement. All the irrational statements are corrected accordingly in the revised manuscript introduction.
Comment #3: Nanomaterials are materials in nano size and AC is a type of carbon materials. There are nano-sized AC as well. If the authors intended to compare nano-sized AC with ZVI or other materials, they need to specify.
Response to the comment: Again, sorry for the confusing statement. All the irrational statements are corrected accordingly in the revised manuscript.
Comment #4: Introduction should have some background about nZVISLP or similar materials and motivation on the synthesis of nZVISLP. There have been several studies on the removal of dyes including methylene blue with nZVI, nZVI fabricated with other materials, biogenic nZVI. The manuscript should address how nZVISLP is different or unique from other studies and why nZVISLP is useful for dye treatment (hypothesis).
Response to the comment: Thank you for suggestive comments that can improve quality of our article. As suggested, various studies where nZVI is synthesized from various biogenic materials are reviewed and the significance & its importance has been included in the revised manuscript at line 58-67.
Comment #5: Figure 1 is not needed since it is a well-known dye and the chemical structure is readily searchable. If the authors intended to discuss about any functional groups for FTIR or sorption affinity, address such points with Figure 1.
Response to the comment: As suggested, Figure 1 has been removed from the revised manuscript.
Comment #6: L100. Add detailed method for SEM procedure including preparation of sample, EDX software,
Response to the comment: As suggested, a detailed method for SEM has been provided in the manuscript. Page no. 3 line no. 104 to 106.
Comment #7: L187. How did the authors prepare samples for SEM? The SEM image shows aggregates or films rather than individual particles. Based on the image, the materials do not appear as nanoparticles rather than materials containing nano-sized pores or particles embedded in a film or carbon tape. It may be due to artifact from sample preparation. The authors intended to observe micromorphology of the nanoparticles, the SEM images are needed to be retaken.
Response to the comment: The sample preparation method for SEM has been added in the manuscript (Page no. 3, Line no.104 to 106). The SEM analysis of nZVISLP has been redone and updated images are included in revised manuscript as Fig. 3a and 3b.
Comment #8: L190-192 The EDX analysis is semi-quantitative unless the sample was prepared in an epoxy perk and polished at the same height. If Table 1 is based on EDX results, the authors should note that. Additionally, detection of oxygen do not mean oxidation of iron since EDX does not indicate oxidation state of elements. Wet-chemical (e.g. aqua regia) digestion with ICP analysis could provide atomic composition accurately.
Response to the comment: Thank you for your suggestive remarks. Unfortunately, ICP is not available in our institute and therefore only EDX analysis of the nZVISLP was performed. This EDX analysis is repeated and the revised results are presented as Figure 3(c) in the revised manuscript. The method for sample preparation for EDX has been added in the page no. 3 line no. 106 to 108.
Comment #9: L197-206 Move this to the method.
Response to the comment: The pHZPC being an essential characteristic of any adsorbent that represents a particular pH value at which the surface charge of that adsorbent is neutral. The pHZPC of the synthesized material was determined at different pH and the outcome was presented in section 3.1.3
Comment #10: L213 Why did the authors use ethanol for dispersion medium for DSL (Zeta nanosizer) analysis?
Response to the comment: Thank you for the remark. The synthesized adsorbent was well dispersed in ethanol having good stability that help to precise measuring of zeta potential (Zeta potential -28 mV).
Comment #11: I suggest to move BET plots (Figure 6) to SI. It is more interesting to compare the surface area with other nZVI or similar nanomaterials used for dye treatment. The authors could calculate surface area based particle size measured by Zeta nanosizer with the assumption of spherical shape and compare it with BET results, which may indicate porous structure of nZVISLP if BET surface area is greater than the calculated value.
Response to the comment: As suggested, the BET plots have been moved to supplementary material (see Figure S2) and surface area is calculated on the basis of PSA size and the results are presented in revised manuscript. Page no. 8 and 9, line no.226-237.
Comment #12: XRD results. XRD will have limited utility since detection limits aren’t that great, especially when mineral grain size is nano-ish. This result may be due to the amorphous nature of nZVISLP, nanosize or both. It is good to discuss this in the text but Figure 7 could be moved to SI.
Response to the comment: Thank you for your remarks. As suggested Figure 7 has been moved to supplementary material (See Figure S3) of the revised ESM.
Comment #13: Section 3.2. The authors measured disappearance of MB in the presence of nZVISLP. Why did the authors think the disappearance of MB is due to adsorption? Many studies have shown MB reacts with nZVI to form colorless leuco-MB by oxidation/reduction by nZVI, to be precipitated with Fe species, to be degraded by -OH radicals generated by Fe and oxygen along with adsorption. Adsorption could be a dominant removal mechanism of MB by nZVISLP but the authors should discuss other potential removal mechanisms prior to sorption isotherm (section 3)
Response to the comment: The possible mechanisms for MB removal other than adsorption that may the influence dye removal are included in the revised manuscript. Page. No. 11, line no. 311-316.
Comment #14: L267-277 The maximum adsorption capacity does not seem to be different with increase of C0. What is standard error or standard deviation for replicated results of each experiment? Is the difference statistically significant?
Response to the comment: Sorry for the typo error. qt was replaced with qmax in the revised manuscript (page no. 11, line no. 296). As suggested, the standard deviation for each result has been included in the revised manuscript.
Comment #15: Figure 8. It is more insightful to plot concentration of MB over time or ln C/C0 of MB over time rather than removal rate over time. Plotting concentration of lnC/C0 of MB over time allows us to reaction rate order, constant, or potential mechanisms. Figure S1 should be in the manuscript.
Response to the comment: As suggested by reviewers and editor to limit the figures, Figure S1 is still retained in ESM, but the information and significant outcomes are expressed in the text. Figure 5 (old Figure 8) is redrawn in the revised manuscript to present the MB residual concentration vs time for the process parameters.
Comment #16: Section 3.2.2 Did the author measure pH during or after the experiment? It does not appear the authors used buffer solution such as carbonate buffer. Changes in pH is common with the presence of nZVI or other sorbent. Figure 8b shows removal percentage increased with time for pH 4, 6, and 7.
Response to the comment: The pH of the solution was checked after adsorption for every time interval, it was noted that pH was changing and the final pH of the solution was noticed to near it pHZPC.
Comment #17: Section 3.4 Predicting kinetic from experimental sorption measured in this specific system is important from a practical point of view. However, I am concerned about kinetic modeling since the authors did not clearly address boundary conditions and assumptions. Reaction kinetics is not simple finding reaction order or linear/non-linear fitting but shed a light on reaction mechanisms. Pseudo reaction order is only valid when one of the reactants is in excess or relatively constant. When the authors conducted kinetic calculation of MB, reaction sites of nZVISLP should not be limited for calculating pseudo reaction rate constant. The authors could use initial reaction rate (before 50% removal). If the authors intended to get intrinsic values, sorption mechanisms should be proposed first and conduct kinetic simulation and fitting. Below is a review article which may be useful.
Theoretical models of sorption kinetics including a surface reaction mechanism: A review 2009 Advances in Colloid and Interface Science, Vol 152, Issues 1–2, 30 Pages 2-13 A
Response to the comment: Thank you for sharing a good article, and this article presents review of theoretical models that describe the kinetics of pollutant adsorption as well as the behaviour characteristic of physical kinetic processes. Our study is limited to identification of suitable kinetics that can describe the MB adsorption using nZVISLP nanoparticle. We will definitely consider your suggestion in our future publications. Indeed, this is an excellent article to get more insights into sorption process.
Comment #18: Define variable and parameters of equation in Table and figures.
Response to the comment: As suggested all the variables and parameters of equations shown in Table and figures are defined and indicated accordingly in the text or below tables or in figures.
Reviewer 2 Report
The title suggests an interesting paper, but it is misleading, as in my opinion no ZVI has been produced which questions the entire research. Therefore I suggest to reject the paper.
Detailed comments:
Line 2: replace "particle" by "particles"
The language of the article needs to be significantly improved. Already the first sentence servers as a good example: "AS THE presence of dyes in water bodies posES severe problemes FOR humanS and aquatic creatures, treatment methdos for removal of these polutants from water bodies is OF utmost importance". In this sentence also "aqueous water bodies" is a pleonasm.
line 24: Rephrase! evolutionary...evolution... The sentence is not understandable. to find or to found?
line 28: "isotherms which describe MB adsorption". "vigorous" is an adjective, "non-linearity" is a noun, they cannot be combined this way.
line 29: you cannot write "the" non-error functions, as these functions have not been introduced before. Be precise with the use of articles (personal and non-personal articles or no article at all). Why does the non-linearity of the adsorption model uphold the performance of an optimization technique? And what does this technique optimize: a) the mathematical function that describes the adsorption or b) the adsorption itself?
line 38: Not all dyes and all pigments show a mutagenic and toxic effect! Siena, for example is a non-toxic pigment.
line 39: Not all dyes are non-biodegradable and categorized as carcinogenic! Indigo is no cancerogenic, to give one famous example.
line 40: There are too much generalizations. Basis Red 29 is a cationic dye which is not aquatoxic or human-toxic. However, MB is toxic, category 4, and dangerous for the water, category 2.
line 41: "man-made". "are present in effluents..."
Here I will stop the detailed language revision and focus on the main content.
line 50: "biological oxygen demand" is a sum parameter, not a treatment method.
line 52: there are lots of adsorbents, i.e. you should not say "the adsorption technique"
line 57: Do you mean mechanical strength?
line 62. This general statement is contradictory to the still much more abundant use of AC in waste water treatment compared to nanoparticles which are still a niche application.
In general the introduction is far too general and should focus on nano-ZVI. How is it produced (explain the relation to sweet lime)? Give an overview about studies using ZVI for dye removal! Explain the novelty of your research compared to those studies.
line 79: replace "identified" by "produced"
General remark on the nZVISLP synthesis: I doubt that you produced ZVI by this process. When you increase the pH of a FeSO4 solution (may it be with or without sweet lime pulp) by NaOH, FeOOH should precipitate. There is no reason why Fe(III) should be reduced to Fe(0).
FTIR is not a suitable method to show that really Fe(0) has formed. SEM analyses indicate that it is not ZVI, but a mixture of various compounds. XRD evaluation is highly doubtful. The description of the "green ZVI" reminds me of "green rust".
I stop my review here and suggest to reject the paper as the principial assumption, i.e. the use of ZVI, is not correct.
Author Response
We appreciate the valuable comments of all the reviewers on our manuscript and express our sincere thanks for their time and constructive suggestions. The followings are the justifications presented in reply to reviewers’ comments. The critical comments and useful suggestions have been helped us to improve our paper considerably. As indicated in the reply’s that follows, we have taken these comments and suggestions into account, and amended/corrected the statements in the revised version of our manuscript which are with text with yellow background color in the revised manuscript.
Response to reviewer 2 comments:
The title suggests an interesting paper, but it is misleading, as in my opinion no ZVI has been produced which questions the entire research. Therefore, I suggest to reject the paper.
Response to the comment: Based on your valuable comments as well as the other two reviewer’s suggestions, the manuscript is completely overhauled. I hope the revised manuscript appeals to you and reconsiders your decision.
Detailed comments:
Comment #1:
Line 2: replace "particle" by "particles"
Response to the comment: The suggested modification has been done. Page no.1 line no.2. Thank you.
Comment #2: The language of the article needs to be significantly improved. Already the first sentence servers as a good example: "AS THE presence of dyes in water bodies poses severe problems FOR humans and aquatic creatures, treatment methods for removal of these pollutants from water bodies is OF utmost importance". In this sentence also "aqueous water bodies" is a pleonasm.
Response to the comment: The authors tried to improve the language and overall quality of the manuscript by getting help from a native speaker fluent in technical English.
Comment #3: line 24: Rephrase! Evolutionary...evolution... The sentence is not understandable to find or to found?
Response to the comment: All the ambiguous statements are rephrased in the revised manuscript. Page no1, Line 25.
Comment #4: line 28: "isotherms which describe MB adsorption". "vigorous" is an adjective, "non-linearity" is a noun, they cannot be combined this way.
Response to the comment: Modified as per the reviewer’s comments. Page no. 1, line no. 29.
Comment #5: line 29: you cannot write "the" non-error functions, as these functions have not been introduced before. Be precise with the use of articles (personal and non-personal articles or no article at all). Why does the non-linearity of the adsorption model uphold the performance of an optimization technique? And what does this technique optimize: a) the mathematical function that describes the adsorption or b) the adsorption itself?
Response to the comment: This statement is rephrased in the abstract. Usage of non-linear functions and their significance was explained in section 2.
Comment #6:
line 38: Not all dyes and all pigments show a mutagenic and toxic effect! Siena, for example, is a non-toxic pigment.
line 39: Not all dyes are non-biodegradable and categorized as carcinogenic! Indigo is no cancerogenic, to give one famous example.
Response to the comment: The above contradictory statements have been modified. Page no. 1, line no. 38-39. Thank you.
Comment #7: line 40: There are too much generalizations. Basis Red 29 is a cationic dye which is not aquatoxic or human-toxic. However, MB is toxic, category 4, and dangerous for the water, category 2.
Response to the comment: The above-mentioned statement has been modified. Page no.2, line no. 42. Thank you.
Comment #8: line 41: "man-made". "are present in effluents..."
Response to the comment: The sentence has been rephrased. Page no.2, line no. 42-43.
Here I will stop the detailed language revision and focus on the main content.
Comment #9:
line 50: "biological oxygen demand" is a sum parameter, not a treatment method.
Response to the comment: This part of the sentence has been removed from the revised manuscript.
Comment #10: line 52: there are lots of adsorbents, i.e. you should not say "the adsorption technique"
Response to the comment: The sentence has been modified. Page no. 2, line no. 54.
Comment #11: line 57: Do you mean mechanical strength?
Response to the comment: The word has been modified at page 2, Line 58.
Comment #12: line 62. This general statement is contradictory to the still much more abundant use of AC in waste water treatment compared to nanoparticles which are still a niche application.
Response to the comment: The sentence has been removed from the revised manuscript.
Comment #13: In general, the introduction is far too general and should focus on nano-ZVI. How is it produced (explain the relation to sweet lime)? Give an overview about studies using ZVI for dye removal! Explain the novelty of your research compared to those studies.
Response to the comment: The introduction has been revised and specified. The main objective of this study is to synthesize nanoparticles from waste/biowaste material (sweet lime pulp waste) and investigate its performance for MB removal from aqueous solution. So its application is the novelty in this study. Another significant novelty in this study is the application of a hybrid evolutionary differential evolution optimization technique that has been applied to estimate the optimal intrinsic parameters in the isotherm and kinetic models. This is the first kind of application in this arena.
Comment #14: line 79: replace "identified" by "produced"
Response to the comment: The above mentioned correction has been done. The sentence has been modified. Page no. 3 line no. 82.
Comment #15: General remark on the nZVISLP synthesis: I doubt that you produced ZVI by this process. When you increase the pH of a FeSO4 solution (may it be with or without sweet lime pulp) by NaOH, FeOOH should precipitate. There is no reason why Fe(III) should be reduced to Fe(0).
Response to the comment: The synthesis of nZVISLP was done according to the methods described in the literature by previous researchers (Green synthesis of iron nanoparticle from extract of waste tea: An application for phenol red removal from aqueous solution: https://doi.org/10.1016/j.enmm.2018.08.003; Biogenic reductive preparation of magnetic inverse spinel iron-oxide nanoparticles for the adsorption removal of heavy metals: https://doi.org/10.1016/j.cej.2016.08.067) and the colour transformation from brown to black represents the reduction of Fe+2 to F0 by the biomolecules present in sweet lime pulp extract. And the further discussion has been done in the revised manuscript. Page no. 3, line no. 88-98. Thank you
Comment #16: FTIR is not a suitable method to show that really Fe(0) has formed. SEM analyses indicate that it is not ZVI, but a mixture of various compounds. XRD evaluation is highly doubtful. The description of the "green ZVI" reminds me of "green rust".
Response to the comment: The SEM analysis of the sample was repeated and the images are shown in figure 3. The images show average size of particles was observed 99.5 nm diameter however, particles are still in aggregates form.
Reviewer 3 Report
This research study investigates the elimination of methylene blue (MB) from aqueous solution using zero-valent iron nanoparticles synthesized from sweet lime pulp waste (nZVISLP). There are some suggestions to improve the manuscript as outlined below:
-The novelty of the work needs to be demonstrated, is it in the synthesis method? Application? The concept of application of Iron nanoparticles for environmental remediation is not new.
-Introduction section needs to be written better, Please compare this work with the recent publication Green synthesized nanoclusters of ultra-small zero valent iron nanoparticles as a novel dye removing material.
-please explain why isotherm model has been chosen for this work
-How many replicates were involved, please also reference the methods that have been used from elsewhere
-SEM graph is not clear enough
-XRD needs to be indeed properly
-I don’t see any error bar in Fig 8, is this mean experiment has been performed just once?
-Discussion section needs to be improved by discussing some recent published works in the same area
Author Response
We appreciate the valuable comments of all the reviewers on our manuscript and express our sincere thanks for their time and constructive suggestions. The followings are the justifications presented in reply to reviewers’ comments. The critical comments and useful suggestions have been helped us to improve our paper considerably. As indicated in the reply’s that follows, we have taken these comments and suggestions into account, and amended/corrected the statements in the revised version of our manuscript which are with text with yellow background color in the revised manuscript.
Response to Reviewer 3 comments
This research study investigates the elimination of methylene blue (MB) from aqueous solution using zero-valent iron nanoparticles synthesized from sweet lime pulp waste (nZVISLP). There are some suggestions to improve the manuscript as outlined below:
Comment #1: The novelty of the work needs to be demonstrated, is it in the synthesis method? Application? The concept of application of Iron nanoparticles for environmental remediation is not new.
Response to the comment: Thank you for your remark. Yes, we authors agree with you that the concept of application of Iron nanoparticles for environmental remediation is not new. But in this study, the main objective of this study is to synthesize nanoparticles from waste/biowaste material (sweet lime pulp waste) and investigate its performance for MB removal from aqueous solution. Therefore, its synthesis and application is the novelty in this study. Another significant novelty in this study is the application of a hybrid differential evolution optimization technique that has been applied to estimate the optimal intrinsic parameters in the isotherm and kinetic models. This is the first kind of application in this area.
Comment #2: Introduction section needs to be written better, please compare this work with the recent publication, “Green synthesized nanoclusters of ultra-small zero valent iron nanoparticles as a novel dye removing material”.
Response to the comment: The introduction section has been revised and modified. Above suggested article, synthesized zero-valent iron nanoparticles from Leafy branches of Mediterranean cypress to remove Methyl orange dye. So this study is completely different from the present study.
Comment #3: Please explain why isotherm model has been chosen for this work
Response to the comment: Isotherm models are needed to understand the inherent mechanism involved in the MB adsorption by nZVISLP. This will also provide insights into the nature of interactions between adsorbent and adsorbate.
Comment #4: How many replicates were involved, please also reference the methods that have been used from elsewhere
Response to the comment: All the experiments were run in triplicates and the average value is used in the study. This statement is included in the revised manuscript. Also, the appropriate references for the methods used have been added in the revised manuscript.
Comment #5: SEM graph is not clear enough
Response to the comment: SEM analysis has been redone and the results are updated in Figure 3 of the revised manuscript.
Comment #6: XRD needs to be indeed properly
Response to the comment: As suggested, the XRD analysis is included in the revised manuscript. Page no. 3, line no. 108-110 and results discussed on page no 9, Section 3.1.6.
Comment #7: I don’t see any error bar in Fig 8, is this mean experiment has been performed just once?
Response to the comment: The experiments were performed in triplicates, so the error-bars are added to the plots shown in Figure 5 to represent the deviation in experimental results.
Comment #8: Discussion section needs to be improved by discussing some recent published works in the same area
Response to the comment: Thank you for the suggestion. The discussion section has been improved by the addition of recent articles.
Round 2
Reviewer 2 Report
The principal issues which I clearly explained in my first review have not been solved. Therefore I stay with my recommendation to reject this paper.
Reviewer 3 Report
Authors made the required changes and I suggest acceptance